# Bacterial and Viral Pathogens with One Health Relevance in Invasive Raccoons (*Procyon lotor*, Linné 1758) in Southwest Germany

**DOI:** 10.3390/pathogens12030389

**Published:** 2023-03-01

**Authors:** Nico P. Reinhardt, Judith Köster, Astrid Thomas, Janosch Arnold, Robert Fux, Reinhard K. Straubinger

**Affiliations:** 1Bacteriology and Mycology, Institute for Infectious Diseases and Zoonoses, Department of Veterinary Sciences, LMU Munich, 80539 Munich, Germany; 2Aulendorf State Veterinary Diagnostic Centre (STUA), 88326 Aulendorf, Germany; 3Wildlife Research Unit, Agricultural Centre Baden-Wuerttemberg (LAZBW), 88326 Aulendorf, Germany; 4Virology, Institute for Infectious Diseases and Zoonoses, Department of Veterinary Sciences, LMU Munich, 80539 Munich, Germany

**Keywords:** *Procyon lotor*, One Health, pathogen pollution, Germany, *Leptospira* spp., *Anaplasma phagocytophilum*, carnivore protoparvovirus-1, canine distemper virus, West Nile virus, influenza A virus

## Abstract

In Europe, raccoons are invasive neozoons with their largest population in Germany. Globally, this mesocarnivore acts as a wildlife reservoir for many (non-)zoonotic (re-)emerging pathogens, but very little epidemiological data is available for southwest Germany. This exploratory study aimed to screen free-ranging raccoons in Baden-Wuerttemberg (BW, Germany) for the occurrence of selected pathogens with One Health relevance. Organ tissue and blood samples collected from 102 animals, obtained by hunters in 2019 and 2020, were subsequently analysed for two bacterial and four viral pathogens using a qPCR approach. Single samples were positive for the carnivore protoparvovirus-1 (7.8%, n = 8), canine distemper virus (6.9%, n = 7), pathogenic *Leptospira* spp. (3.9%, n = 4) and *Anaplasma phagocytophilum* (15.7%, n = 16). West Nile virus and influenza A virus were not detected. Due to their invasive behaviour and synanthropic habit, raccoons may increase the risk of infections for wildlife, domestic animals, zoo animals and humans by acting as a link between them. Therefore, further studies should be initiated to evaluate these risks.

## 1. Introduction

Biodiversity loss and the climate crisis weaken the resilience of ecosystems. Additionally, the merging of spatial boundaries between domestic animals, wildlife and humans, as well as microbial and vector adaptation, ultimately results in increasing environmental pollution and the distribution of (non-)zoonotic pathogens, particularly with (re-)emerging infectious diseases (EIDs) [1,2,3,4]. EIDs pose a threat to public health as, according to some publications, the majority of them bear a zoonotic potential (60.3%), and most of which derive from wildlife (71.8%) [1,3,4]. For this reason, it is crucial to address these pathogens and their hosts in the view of a One Health perspective [5,6].

One important host is the raccoon, which is a mesocarnivore belonging to the procyonid family, native to North and Central America and increasingly spreading as an invasive alien species (IAS) in Asia and Europe [7,8]. Raccoons have been included on the Union list of IAS of Regulation (EU) No 1143/2014 Art. 19 since 2016. In Germany, raccoons occur in varying population densities in all 16 federal states. There are two main foci of distribution in central and northeast Germany, but they are expected to populate most of the country by the middle of the 21st century [7,8]. As opportunistic generalists, raccoons live in a wide variety of habitats. Their synanthropic lifestyle is especially important from an epidemiological perspective, because it leads to smaller home ranges and higher population densities due to advantages such as abundant food availability and shelter possibilities [9,10,11]. Raccoon density in the cities of Kassel and Bad Karlshof reaches 90–110 animals/km^2^ [12]. The number of raccoons being hunted has substantially increased with a focus on northeast Baden-Wuerttemberg (BW), where the hunting bag amounted to 339 in the hunting season 2010/11 and increased to 4015 in 2020/21 [13]. Fischer et al. [7] estimate that the hunting bag data reflect about 10% of the actual population density, which would result in a total number of approximately 40,000 animals in BW. There, according to §7 section 8 of the Hunting and Wildlife Management Act (JWMG), the raccoon is under utilisation management, and consequently their spread needs to be counteracted (§5 section 3 JWMG).

Raccoons combine a wide-ranging distribution pattern with a high ecological damage potential. For example, they have negative interactions (competition, predation, disease transmission) with other wildlife and, being a reservoir for a variety of pathogens, also represent a threat to public and domestic animal health [9,14,15,16]. The occurrence of pathogens in raccoons has rarely been investigated in Germany, in particular in its southern regions. Therefore, we conducted a study, in which helminths and selected, partly vector-borne (non-)zoonotic and (re-)emerging viral and bacterial pathogens with One Health relevance, which the raccoon is known to host, were investigated. As a part of that survey, here we have focused on four viruses: the carnivore protoparvovirus-1 (CPPV-1) [17,18,19,20,21], canine distemper virus (CDV) [22,23,24], influenza A (IAV) [25,26,27] and West Nile virus (WNV) [28,29,30]; as well as two bacteria, the pathogenic *Leptospira* spp. (p*L*) [21,31,32] and *Anaplasma phagocytophilum* (*Ap*) [16,33,34]. Using specific qPCR methods, we screened tissue and blood samples of randomly collected free-ranging raccoons in BW for the selected pathogens. This study highlights the occurrence of CDV, CPPV-1, p*L* and *Ap* in raccoons from BW. The raccoons’ synanthropic behaviour as an IAS make investigations of (non-)zoonotic EIDs especially necessary, for which it is susceptible to better understand its role in pathogen pollution and as a link between infections in wild and domestic animals and humans [1,2,4,10,16,34].

## 2. Results

A total of 30 out of 102 raccoons (28.3%, 95% CI: 20.8–39.3) were positive in qPCR of at least one of the detected four pathogens. Positive results were obtained for CPPV-1 (n = 8, 7.8%, 95% CI: 3.4–14.9), CDV (n = 7, 6.9%, 95% CI: 2.8–13.6), p*L* (n = 4, 3.9%, 95% CI: 1.1–9.7) and *Ap* (n = 16, 15.7%, 95% CI: 9.2–24.2) (Table 1 and Figure 1b). Neither IAV in lung tissue nor WNV in brain samples could be detected by RT-qPCR, so both are excluded from Table 1 and Figure 1b. Dual infections were seen in five raccoons. Two animals, both from a rural area, were infected with p*L* and CDV (WB18 (male, adult), WB19 (female, juvenile)). Raccoon WB34 was a female adult raccoon from an urban area with a coinfection with CDV and CPPV-1. Raccoon WB73 (rural, female, juvenile) was positive for *Ap* and p*L* and one male, urban, adult raccoon (WB91) was positive with *Ap* and CPPV-1. Interestingly, all five coinfections originate from OAK and SHA, which are neighbouring counties (Table 2 and Figure 1b).

The classification into urban and rural sampling communities showed that 66.7% (68/102) of the raccoons originated from urban areas and 33.3% (34/102) from rural areas (Table 1 and Figure 1a). In urban areas, 10.3% (7/68) were hunted on private property. Raccoon WB90 was hunted in the municipality of Essingen (OAK), partly urban according to the Federal Office for Building and Regional Planning (BBSR) [35,36], but since the hunting ground is known to be in a rural area, it was also categorised as rural.

At the time of necropsy, 80.4% (82/102) of the animals examined were classified as fresh, 12.7% (13/102) as minimally and 4.9% (5/102) as moderately autolysed. None of the animals collected was severely autolysed. Two raccoon torsos (1.96 %) were mutilated; one intestine and one liver were incomplete due to hunting. In three of the hunted raccoons, the hunters described behavioural abnormalities such as aggressiveness (WB02), ataxia/apathy (WB19) and loss of shyness (WB23).

### 2.1. Carnivore Protoparvovirus-1

In eight intestine samples, CPPV-1 was detected with Ct-values between 23.2 and 37.6 by qPCR [37]. Five of the CPPV-1-positive animals originated from OAK, two from RMK and one from HD (Figure 1b). Raccoon WB91 (FPV_VP2_Raccoon_Germany), male and adult, with a Ct-value of 23.2, had a dual infection with *Ap* (Ct = 29.51) and came from Wasseralfingen in the municipality of Aalen (OAK). The raccoon WB34 was also tested positive for CDV (Table 2). There were no pathohistological results of the intestine in WB91. All CPPV-1 positive raccoons came from urban areas (Table 1).

The genotype of one intestinal sample positive for CPPV-1 (FPV_VP2_Raccoon_Germany) could be assigned to the FPV-clade (Figure 2). We analysed the first 647 (amino-terminal) nucleotides (nt) of the VP2-gene and detected an amino acid pattern typical for FPV at aa80: lysine; aa93: lysine; and aa103: valine.

### 2.2. Canine Distemper Virus

In seven samples out of 102 (6.9%), CDV was detected with a RT-qPCR from brain tissue [38]. Ct-values in the positive samples ranged from 18.4 to 37.4. The positive animals could be assigned to the districts of SHA (n = 3), RMK (n = 2), HD (n = 1) and OAK (n = 1; Figure 1b). All three raccoons from SHA (WB18, WB19, WB93) came from the municipality of Frankenhardt, and in two of them, coinfections with p*L* were detected (Table 2). With a Ct-value of 18.4, raccoon WB19, a juvenile female raccoon from Frankenhardt (SHA), was described by the hunter as ataxic and lethargic, had a fasting gastrointestinal tract at necropsy and was classified as moderately emaciated (weight: 1.2 kg). Pathohistological examinations of the brain, lungs and liver showed focal submeningeal haemorrhage, minor alveolar pulmonary oedema and a pyogranuloma in the liver. Raccoon WB19 had coinfection with p*L* (Ct = 31.2). The adult, female raccoon WB34 from Aalen (OAK) with a Ct = 28.5 had a coinfection with CPPV-1 (Ct = 36.9; Table 2).

### 2.3. Pathogenic Leptospira *spp.*

Using qPCR, which was carried out according to the Adiavet Lepto Real Time Kit (Bio-X, Adiagene, Rochefort, Belgium), specific fragments of the genome of the seven p*L* strains were detected in the liver and kidney samples of four of the 102 raccoons. Three were considered positive (WB08 Ct = 33.3, WB18 Ct = 37.4, WB19 Ct = 31.2), and one raccoon (WB73) was weak positive with a Ct-value of 40.1. Of those four, one came from KA and three from SHA, where three coinfections were detected (WB18, WB19, WB73; Table 2 and Figure 1b). The pathohistological findings for the raccoon WB19 have already been described in Section 2.2. The extractions of the four positive samples from the liver and kidney were re-examined with a different PCR method according to Ferreira et al. [39] to verify them before being sent to Boehringer Ingelheim Animal Health France (Lyon). The initially weak positive raccoon (WB73) could not be confirmed as positive. Ct-values of the positive kidneys (WB08 Ct = 28.09, WB18 Ct = 32.58, WB19 Ct = 26.24) were slightly lower than those from the livers (WB08 Ct = 37.71, WB18 Ct = 34.34, WB19 Ct = 32.81). The confirmed DNA extractions from the three positive raccoons with Ct < 40 were then sent to Boehringer Ingelheim Animal Health France (Lyon) for molecular genetic differentiation of the *Leptospira* serogroups. The detection of p*L* was confirmed in all three samples, and the genetic differentiation revealed in all three kidney extractions the *Leptospira interrogans* serogroup Australis and in raccoon WB08 and WB18 also serogroup Icterohaemorrhagiae; WB19 was only positive for serogroup Australis. In the liver samples, only WB18 was positive for *L. interrogans* serogroup Icterohaemorrhagiae.

### 2.4. Anaplasma phagocytophilum

Blood and spleen samples of 16 of 102 raccoons tested positive for the *msp2*-gene of *Ap* using a specific qPCR test [40]. Of the 16 samples, 12 were positive for *Ap* in both spleen and blood. Four of the 16 samples were positive in blood only. The Ct-values of the blood samples ranged from 26.6 to 35.2, and those of the spleen samples from 26.4 to 34.4. In all but one (WB74) of the blood samples (mean Ct = 29.59), Ct-values were lower than in the spleen samples (mean Ct = 31.11). *Ap*-positive raccoons originated from OAK (n = 7), SHA (n = 5), GÖ (n = 2) and HN (n = 2) (Figure 1b). Therefore, 75% of the *Ap*-positive raccoons came from OAK and SHA. A significant (*p* ≤ 0.025) relationship (*p* = 0.01, OR = 0.24; CI: 0.06–0.81)) was observed between a positive result and the area category with Fisher’s exact test; therefore, 10 of 16 *Ap*-positive raccoons came from rural areas (Table 1 and Figure 1b). No other significant dependencies could be determined with Fisher’s exact test. Using the directional MWU test, we found that raccoons positive for *Ap* (weight: 5.1 kg, length: 67 cm) were significantly heavier and larger (weight: *p* = 0.025, length: *p* = 0.004, significance level *p* ≤ 0.05) than negative raccoons (weight: 4.3 kg, length: 57 cm).

## 3. Discussion

In this study, we found pathogens with One Health relevance in free-ranging raccoons in Baden-Wuerttemberg (BW, Germany). In total, 30 out of 102 raccoons were positive in PCR for at least one of the four detected pathogens. There were positive results for CPPV-1, CDV, p*L* and *Ap* (Table 1 and Figure 1b). Based on these results, the raccoons’ tendency to spread and their synanthropic habits, raccoons may contribute to the environmental pathogen pollution. Therefore, raccoons may increase the risk of infection for wildlife, domestic animals, captive zoo animals and humans by acting as a link between these groups [1,4,7,16,34].

As described in the results, we detected five dual infections that all originated in SHA and OAK counties. This can be explained by a relatively high raccoon density in these areas and, thus, the high risk of infection with specific pathogens, such as *Ap* and CDV (Figure 1b). Coinfections associated with immunosuppression triggered by CDV and *Ap* infections have already been described in the literature, also regarding CDV in raccoons [41,42,43,44].

### 3.1. Method Limitations

The focus of the raccoon acquisition was on the northern half of BW, as this is where the population density and, thus, the hunting numbers are highest (Figure 1). It should be noted that the spatial and temporal raccoon carcass distribution and submission are subject to bias, as it depends on many complex biological (e.g., raccoon population dynamics) and human-related factors (e.g., the commitment of hunters in the respective areas) [45]. Since the present study aimed to test for the presence of specific pathogens in the BW raccoon population, we limited our sample size to about 100 animals. To acquire precise prevalence and population estimates, the numbers would have to be higher.

As already explained, with a hunting bag of 4015 raccoons during hunting season 2019/2020, the raccoon population was estimated at about 40,000 animals in BW [8,13]. A sample of 102 raccoons therefore represents about 3.3% of the hunting bag and 0.3% of the estimated population in BW. This sample size allows only limited conclusions to be drawn from the actual population structure. The fact that two-thirds of the sampled raccoons originate from urban areas (proportion of urbanised areas of at least 50%) is to be expected considering the dense settlement of the sample area and the higher raccoon densities in urban areas (Figure 1a) [9,11,35]. The proximity of the raccoon’s habitats to human settlements is also evident from the fact that 10.3% of the urban raccoons were collected on private property. However, the division into urban and rural is just a tool, as only the hunting community was categorised, but not the spatial use of the individual raccoon. Home range sizes can vary depending on the season, habitat type, area type (urban > rural), sex (male > female) and age (adult > juvenile) [9,14,21].

The age categorisation based mainly on weight is particularly inaccurate, which is why other parameters were also considered. The body weight of raccoons can vary considerably by season depending on the latitude, as they accumulate a thick fat depot during the winter months [46].

It is probable, despite the high sensitivity and specificity of the established PCR methods, that the individual pathogens’ prevalences are underrepresented. Factors such as autolysis, tissue sample selection and sampling timing, which depend on pathogenesis and incubation time, may have influenced the detection rate [31,38,40,47,48,49,50].

### 3.2. Carnivores Protoparvovirus-1

The CPPV-1 prevalence of 7.8% is within the range of previously published data in carnivores in Germany. In Europe and Germany, there is only one other published case of CPPV-1 in raccoons from Berlin with a prevalence of 7.5% (N = 40), associated haemorrhagic enteritis in the only juvenile, and no apparent intestinal lesions in the two adult raccoons [21]. They did not perform any phylogenetic differentiation. In free-ranging carnivores in Germany, CPV-2-like strains were detected with seroprevalences up to 13% (N = 500) in red foxes (*Vulpes vulpes*) and 6% in raccoon dogs (*Nyctereutes procyonoides*; N = 33) [51]. We did not observe typical intestinal changes such as haemorrhagic enteritis or congestion in any of the positive animals in gross necropsy or in raccoon WB91 in pathohistology [21,52].

In one of the positive animals a phylogenetic analysis was performed (WB91, FPV_VP2_Raccoon_Germany) and we were able to analyse the first 647 (amino-terminal) nucleotides of the VP2-gene, where we detected amino acid substitutions typical for FPV [17,20,53]. The VP2-gene is the dominant structural protein and the main factor affecting the host range and virus–host interactions [53]. Our phylogenetic evaluation showed that the discovered sequence FPV_VP2_Raccoon_Germany is almost identical to a sequence described in 2010 in a raccoon from Canada (GenBank MF069446; Figure 2) [18]. Canuti et al. [18] found that the isolated FPV-sequences, including MF069446, most closely matched isolates from domestic cats, other felines and one raccoon from the USA. Therefore, they have suggested that multiple viral strains of FPV must circulate in North America’s wildlife. In German cats, FPV-infections account for up to 95% of CPPV-1 infections [20]. CPV-2-like variants (CPV-2, CPV-2a), which are more commonly associated with dogs, were detected more frequently in raccoons in North America compared to FPV [18,19,20].

Epidemiologically, it is interesting that all positive raccoons came from urban areas (Table 1). We conclude that the density of domestic animals increases with the human population density. Therefore, urban areas raise the chances of contact with raccoons and cross-species transmission in both directions [54].

### 3.3. Canine Distemper Virus

The calculated prevalence of 6.9% was lower than in previous studies in Germany, despite the poor comparability due to the use of different methods. In this study, the frontal cortex and brainstem were examined, as the virus can persist there for up to three months, depending on the humoral immune defence, but acute infections could be missed [44,47]. However, acute infections cannot be detected with this tissue material, which is why the actual prevalence may have been higher. In Mecklenburg-Western Pomerania (Germany), CDV (European wildlife lineage) was detected for the first time in Europe in 2007 in 22.7% (N = 22) of the raccoons examined [55,56]. One year later, a prevalence of 14.6% (N = 206) was observed in Lower Saxony [57]. In the first major CDV outbreak among free-ranging raccoons in Europe, 74/97 (76.3%) raccoons tested positive for CDV in Berlin in 2012 and 2013 [24]. Consecutive phylogenetic studies revealed that the isolated strains belong to the Europe lineage of CDV and are closely related to those of German red foxes and a domestic dog from Hungary [24].

The detection of positive raccoons from different parts of northern BW suggests a widespread endemic CDV situation. This is also evident from the data of the Chemisches und Veterinäruntersuchungsamt (CVUA) Stuttgart and the State Rabies and Epidemiology Centre Freiburg, which examine raccoons for CDV that have been sent in with behavioural or pathological abnormalities [58,59]. In 2015, the first positive raccoon was detected in LB. Since then, the proportion of examined raccoons and the detection rate of CDV increased. In 2018, 10 positive raccoons were observed; two years later, there were 28 positive, and in 2021, there were already 46 positive raccoons observed in BW [58,59]. There has been a clear epidemiological focus on the districts with a high raccoon density (Stuttgart, Esslingen, GÖ, LB, RMK, OAK, SHA) [58,59]. It is conjectured that the positive raccoons were infected by red foxes or other susceptible animals, as CDV is highly transmissible between species, and the endemic areas of the sensitive species overlap in time and space [24,57,58,59]. Thus, raccoons could be part of a “metareservoir” in BW where CDV infections can be maintained by interacting wildlife reservoir populations [23].

In raccoon WB19, a CDV infection is probably the causative agent due to the low Ct value of 18.3 and the observed characteristic neurological clinical symptoms (ataxia, apathy) [43]. No inclusion bodies were found in the brain histology, which, however, do not always have to be associated with a positive PCR detection [47].

### 3.4. Pathogenic Leptospira *spp.*

To our knowledge, this is the third detection of p*L* in raccoons in Germany and Europe. With a prevalence of 3.9% it lies between the first study published by Anheyer-Behmenburg [57] with a prevalence of 1.3% (N = 457, Lower Saxony) and the second by Rentería-Solís [21] with a prevalence of 12.3% (N = 65, Berlin, Mecklenburg-Western Pomerania).

One reason may be that the weak positive-reacting raccoon was not verified with a second PCR method, as described by Ferreira et al. [39], which might have been less sensitive. Another reason might be that this method only identifies four pathogenic *Leptospira* species (*Leptospira interrogans*, *L. kirschneri*, *L. borgpetersenii* and *L. noguchi*), and not seven as with the Adiavet’s test kit (*L. interrogans*, *L. kirschneri*, *L. borgpetersenii*, *L. nugochii*, *L. weilii*, *L. santarosai*, *L. inadai*). Thus, WB73 might have been weak-positive for *L. weilii*, *L. santarosai* or *L. inadai*. The PCR assay according to Ferreira et al. [39] appeared to be more sensitive in the kidney samples of the three positive raccoons than in the liver samples, as the former showed lower Ct-values.

The PCR method used in this study revealed coinfections in two raccoons with the serogroup Icterohaemorrhagiae and Australis, while the third was only infected with serogroup Australis. Coinfections with multiple serovars have been described more frequently in raccoons, but according to our research, this is the first time in German raccoons, so it could mean a new epidemiological relevance for them as an amplifying host [54]. Rentería-Solís [21] also identified *L*. *interrogans* serogroup Icterohaemorrhagiae (n = 4, Berlin) and Australis (n = 1, Mecklenburg-Western Pomerania). These documented serogroups are highly virulent and can potentially cause severe infections with various clinical signs in animals and humans [60]. In Europe, serogroup Icterohaemorrhagiae is mainly adapted to rats (*Rattus norvegicus*) and is considered the most common serogroup in Germany, causing 35% of human infections. In comparison, serogroup Australis is responsible for 10% of human cases, with hedgehogs serving as the main host [60]. The prevalence of p*L* and individual serovars in wildlife strongly depends on the season, area type, host abundance, fauna biodiversity and other biotic and abiotic factors [50].

From Japan but primarily from North America, there are many records of p*L*, mainly *L. interrogans*, with prevalences of up to 54.8% with a wide range of serogroups having been detected [16,31,32,61]. Raccoons are maintenance hosts which predestines them as sentinels for monitoring epidemiological changes in p*L* and for designating nationwide and regional high- and low-risk regions [31,32,61,62].

### 3.5. Anaplasma phagocytophilum

According to our research, this is the first detection of *Ap* in raccoons in Germany, with 16 of 102 raccoons (15.69%) testing positive. In subclinically and persistently infected animals, the number of circulating organisms varies intermittently, which is why blood and spleen samples were examined [40,48]. The blood samples seem to be more yielding because Ct-values were lower in all but one blood sample.

There are only a few publications on the role of raccoons in the *Ap* infection cycle in Europe [33,34]. With a prevalence of 1.3% (N = 118), which is markedly lower than what we detected, Hildebrand et al. [34] provided the first molecular detection of *Ap* ecotype I in the European raccoon population from Poland. In comparison, the prevalence of *Ap* in red foxes from Germany has been 8.2% (N = 122), and in neighbouring countries, between 2.7% (N = 111) and 34.48% (N = 29) [33,49]. Four ecotypes of *Ap* occur in Europe, with ecotype I being the most important as it is widely distributed and appears in a broad range of hosts, including humans [63,64]. *Ixodes ricinus* is the primary vector of *Ap* in Europe, with prevalences of less than 1% to 17.4% in Germany and 4.5% in BW [65,66,67]. We assume that the positive raccoons were infected with *Ap* by *Ixodes ricinus*.

Raccoons and other (meso-)carnivores can serve as sentinels for the spread of *Ap* in the tick population and can hence be considered indicators of the infection risk for domestic animals and humans in the studied areas [33,41,63,64,65]. Notably, 75% of the *Ap*-positive raccoons came from two districts only (OAK, SHA). Within these districts, however, the distribution is relatively broad. Clusters of three positive raccoons were documented in the municipality of Ellenberg (OAK) and two from Aalen (OAK) (Figure 1b).

In the USA, raccoons have already proven to be an important reservoir and amplifier for *Ap* as they are highly susceptible to the pathogen, can transmit *Ap* to competent tick stages and showed a molecular prevalence of 24.6% [16,33]. Therefore, they could act as propagation hosts, like red foxes, between sylvatic and synanthropic habitats, as they introduce infected ticks into the environment of domestic animals or humans [33,49]. This may lead to increased infestation with and infection of ticks, thus contributing to the pathogenic pollution of *Ap* and other tick-borne pathogens [2,33,42].

We found that raccoons positive for *Ap* were significantly heavier and larger than negative individuals. Most *Ap*-positive animals were adults (12/16; 9/12 males) and males (11/16), both heavier than juveniles or females. Observations in raccoons and canids are consistent with this, where adults are more likely to explore larger geographical areas and consequently are more exposed to higher tick infestations [33,68]. Knowing that *Ap* causes persistent infections, it is reasonable to presume that adults are more likely to carry an infection with *Ap* due to their age. The prevalence rates depend on habitat structure and on the occurrence and availability of typical reservoir hosts or vectors [42,49]. This explains the significant relationship between a positive result and area type; 10 of 16 *Ap*- positive raccoons came from rural areas (OAK, SHA), where the likelihood of being exposed to ticks and thus infected with *Ap* is higher than in urban areas (Table 1 and Figure 1b).

### 3.6. Lack of Detection for IAV and WNV

The IAV and WNV assays were performed using an approved and accredited RT-qPCR protocol with described organ samples. None of the two viruses were detected. It should be emphasised that the sample size was too small to extrapolate this statement to the whole BW raccoon population. Both epidemiological and methodical factors influence the test result and can lead to detection failure. An epidemiological factor is that there has been no transmission of the pathogen into the susceptible population under investigation due to the absence of or insufficient opportunities for infection [25,26,27,30]. Thus, there was no major IAV outbreak in Germany, nor was WNV detected in animals or humans during the sampling period in the sampling region [69,70,71,72,73].

Methodological factors, such as method selection and sample material, which are inconsistent in the few existing molecular studies on IAV and WNV in raccoons, need to be considered. The sample material selected for both IAV (lungs) and WNV (brain) may not be the most yielding tissues, as they only become infected at an advanced stage of infection [26,27]. In raccoons, oral, nasal and faecal shedding is described for both IAV and WNV, which is why oral, nasal and rectal swabs or intestine samples should also be considered as sample probes [25,27,30,74]. The timing of sampling during infection is important; nevertheless, little is known about the pathogenesis of WNV and IAV in raccoons [26,30]. When selecting methods, serological diagnostics should be considered as well for surveillance studies, as IAV- and WNV-exposure might be detected months after infection [28,29,30].

### 3.7. Raccoons and Pathogen Pollution

Besides most anthropogenic environmental factors, host- and pathogen-specific factors influence environmental pathogen pollution and epidemiologic dynamics between wildlife species, domestic animals and humans [1]. Changes in ecosystems, in particular human-caused alterations such as urbanisation and climate crisis, may influence the occurrence and spread of pathogens, vectors or reservoir hosts such as raccoons and thus the epidemiology of (non-)zoonotic EIDs [1,2,5,6,16]. Depending on the interacting species and their ecology, the epidemiology of pathogens (*Ap*, p*L*) may vary in terms of their geographical distribution, including specific variants, abundance and host range [1,33,66]. Accordingly, the raccoon may possibly impact the epidemiology of these pathogens in BW [7,13]. Furthermore, raccoons may help pathogens to persist in the environment for a more extended period, giving rise to new and occasionally more virulent variants of pathogens and contributing to epidemic outbreaks with sometimes high mortality events (CDV, CPPV-1) [17,20,22,23,31,32,44,61].

## 4. Materials and Methods

### 4.1. Sample Collection

In this study, tissue samples from 102 free-ranging raccoons were analysed. The animals were randomly collected between May 2019 and November 2020 in 27 municipalities of 12 counties in BW: City of Heidelberg (HD) and Mannheim (MA) and following districts Ostalbkreis (OAK; Ellenberg, Westhausen, Aalen, Lorch, Essingen, Schwäbisch Gmünd), Schwäbisch Hall (SHA; Fichtenau, Frankenhardt, Stimpfach, Bühlerzell, Kreßberg, Mainhardt), Rems-Murr-Kreis (RMK; Alfdorf, Plüderhausen), Karlsruhe (KA; Rheinstetten), Ludwigsburg (LB; Remseck am Neckar), Main-Tauber-Kreis (MTK; Weikersheim), Heilbronn (HN; Weinsberg, Wüstenrot), Rhein-Neckar-Kreis (RNK; Schriesheim, Dossenheim, Neckargemünd), Rastatt (RA; Durmersheim), Göppingen (GÖ; Börtlingen, Birenbach). More than 10 raccoons each were collected from four counties (HD, OAK, SHA, RMK), between 5–9 from KA as well as LB and less than five animals each from MTK, GÖ, HN, RNK, MA and RA. Until necropsy, the carcasses were stored at −20 °C. No ethical approval was required for this study.

Since the hunting season (according to §10 DVO JWMG 2015, of 25.02.2018) in the sample years for raccoons in BW lasts from August to February, all raccoons were hunted in these months except for one road-killed raccoon in May. In 2019, 32.4% (33/102) of the raccoons were collected from August to December, while in 2020, 67.6% (69/102) of them were collected from January to February and from August to October. Two raccoons were found dead; 100 out of 102 animals were either shot by hunters from a stand (31/100) or caught by a trap and then shot (69/100), according to the current hunting law in BW (§ 41 Section 2 JWMG). The hunters were asked to freeze the collected animals at −20 °C as soon as possible. The municipalities where the hunting sites are located were categorised into rural and urban areas based on the classification of the BBSR [35,36].

### 4.2. Necropsy

One day before the gross necropsy, executed in the BSL3-laboratory of the Parasitology Unit at the University of Hohenheim under the required safety standards, the carcasses were thawed at 20 °C. To assess the quality of the carcasses, they were classified as fresh, minimally or moderately autolytic. Heavily autolytic animals were excluded.

The weight, length and sex of the animals were recorded, and their age was rated as adult (>1-year-old) or juvenile (<1-year-old) based on tooth age eruption and wear, weight and length, reproductive status and hunting history [46,75,76]. Organ samples were collected for molecular studies at necropsy and then rapidly frozen at −80 °C. The tissue samples analysed in this study are listed in Table 3. Brain samples of individuals with behavioural abnormalities described by hunters and visually altered organs were placed in formalin 10% and forwarded to the Aulendorf State Veterinary Diagnostic Centre (STUA, Germany) for pathohistological examination.

### 4.3. RNA and DNA Extraction

At STUA, RNA was extracted from 25 mg of the brain stem and frontal cortex for molecular examinations of WNV and CDV as was the same amount of lung tissue for IAV analyses. DNA was isolated from 25 mg liver and kidney tissue to be tested with qPCR for p*L*. The DNA or RNA from the respective tissue was extracted according to the manufacturer’s instructions of the IndiSpin^®^ Pathogen Kit (Indical Bioscience, Leipzig, Germany). Before extracting the RNA, 10 µL of IC RNA (Indical Bioscience, Leipzig, Germany) was added to the sample as an amplification control.

At the Institute for Infectious Diseases and Zoonoses of the LMU Munich, the isolation of DNA from blood, spleen and intestine tissue was carried out according to the manufacturer’s instruction of the QIAamp DNA Mini Kit^®^ (Qiagen, Hilden, Germany) with a final elution volume of 100 µL. One DNA isolation control (DIC) added with PBS buffer instead of sample material was performed per DNA extraction run. DNA was extracted from blood (200 µL) and spleen tissue (~0.5 mm^3^ in volume) for *Ap*-PCR and from intestinal tissue of the same size for CPPV-1-PCR. The manufacturer’s extraction protocol differed for DNA isolation of blood compared to tissue and was additionally adapted as follows. To mechanically digest the tissue samples, they were placed in a Lysing Matrix Tube A (MP Biomedicals) and mixed with 120 µL (not 100 µL) of PBS buffer, then homogenised using FastPrep-24™ 5G (MP Biomedicals FastPrep^®^ Family) for 40 s at a rotational speed of 6.0 m/s.

### 4.4. PCR Analysis

Every PCR performed at STUA followed accredited standard operating procedures (SOPs). For molecular detection of an 83-bp long viral RNA fragment within the N-protein gene specific for CDV, a RT-qPCR from the brain tissue (brain stem and frontal cortex) was performed according to the previously described protocol of Elia et al. [38] (Table 3). With this method up to 10^2^ copies of RNA can be detected, which indicates a high sensitivity [38]. The same brain samples were tested for the presence of WNV using RT-qPCR according to the method of the Friedrich-Loeffler Institute (FLI) [78] (Table 3). Using primers encoding the 5′-untranslated region (UTR) and the non-structural region NS2A, a differentiation of lineages 1 and 2 of WNV is possible and this sensitive protocol is able to detect 1.1–11 copies of RNA per reaction [78]. The detection of the IAV-genome from the lung tissue by RT qPCR was carried out with the Virotype^®^ Influenza A Kit (Indical; Table 3). We used the qPCR according to the Adiavet Lepto Real Time Kit^®^ (Bio-X, Adiagene, Rochefort, Belgium) to detect specific fragments of the genome (hemolysis-associated Protein 1 (hap-1)) of the seven p*L* species (*L. interrogans*, *L. borgpetersenii*, *L. weilii*, *L. nugochii*, *L. santarosai*, *L. inadai* and *L. kirschneri*) in the liver and kidney samples (Table 3). Differentiation between the individual species is not possible with this PCR. Therefore, in the case of positive p*L* results, DNA was extracted from the positive kidney and liver samples as described above for the tissue samples at the Institute for Infectious Diseases and Zoonoses of LMU Munich. To verify the DNA extractions, a qPCR, according to Ferreira et al. [39], was performed before sending them to Boehringer Ingelheim Animal Health France (Lyon; Table 3). The sensitivity of this PCR is estimated by Ferreira et al. [39] to be <10 genome equivalents (GE). There, a PCR procedure was used to genetically differentiate the individual p*L* samples in eight serogroups (Icterohaemorrhagiae, Australis, Canicola, Autumalis, Pomona, Sejroe, Grippotyphosa, Pyrogenes).

At the Institute for Infectious Diseases and Zoonoses of the LMU Munich, blood and spleen samples were tested for the msp2-gene (77-bp fragment) of *Ap* by qPCR, according to Courtney et al. [40] (Table 3). The sensitivity of this multiplex assay for *Ap* has been shown to be 0.125 infected cells, which is equivalent to approximately <2 copies of single-copy 16S rRNA [40]. In addition, intestinal samples were examined for the VP2-gene (83-bp fragment) of the CPPV-1 to detect FPV and CPV-2 subtypes (CPV-2; CPV-2a, -2b, -2c) by qPCR according to Decaro et al. [37] (Table 3). This method provides precise quantitation between 10^2^ and 10^9^ copies of standard DNA, which is why it is considered highly sensitive [37]. We adapted both PCR protocols to the Qiagen QuantiNova^TM^ Taq polymerase and performed a triplicate approach for each sample on the LightCycler480^®^ system (Roche). The detection format was a monocolour hydrolysis probe with a FAM-adapted fluorescence channel between 465–510nm. The results were evaluated using the Abs Quant/2nd Derivative Max setting. For samples with a Ct-value ≥ 35 (*Ap*) or ≥37 (CPPV-1), the Abs Quant/Fit Points method, in the default setting, was used to calculate the exact Ct-value.

The starting material for the PCR was the extracted RNA or DNA dissolved in a buffer solution or water. Frozen RNA or DNA was thawed shortly before use. The master mix was prepared according to the manufacturer’s or author’s instructions and was transferred to a PCR reaction tube or 96-well plate. Finally, in a separate room, 5.0 µL (CDV, IAV, WNV, p*L*, CPPV-1) or 2.5 µL (*Ap*) samples of DNA or RNA isolate and the controls were added. Before the reaction tubes or 96-well plates were placed in the qPCR device, a short mixing and centrifugation step took place.

The RT-/qPCR evaluation was based on the Ct-values obtained in the FAM channel. Samples with a Ct-value of ≤39 were evaluated as positive, with a Ct-value of >39 as weak positives. The sample was negative if no fluorescence was measurable in the FAM channel. In a triplicate approach, a sample was positive if at least two of three were positive. A triplicate approach was repeated if one of the three approaches was positive. If only one of the three approaches was positive again, this sample was considered negative.

### 4.5. Controls

The correctness of the results was verified using several controls. At least one positive control (PC) was included in each PCR run. Successful amplification was indicated by reaching the predefined Ct-value in the FAM channel. For the *Ap* and CPPV-1 PCR assay, we used in-house PCs with preset Ct-values of 26. The PC for CDV was contributed by CVUA Freiburg, the PC for WNV by FLI and the PC for IAV and p*L*-PCR were included in the test kits. In all PCR runs, the PC must have a Ct-value in the FAM and HEX channels of <35 (according to the manufacturer’s instructions). At least one RNA or DNA isolation control (RIC, DIC) and one negative control (NTC) with nucleic acid-free water were performed per PCR run. Furthermore, a master mix control (MMC), in which the master mix was used as a sample, was added to each of the *Ap* and CPPV-1 PCR assays. If no Ct-value was observed in the RIC/DIC, NTC and MMC, cross-contamination was ruled out in the individual steps. In the RNA extraction procedures at STUA, an amplification control IC RNA was added to every sample. If a PCR test shows a positive result, it is not necessary for the amplification control to be positive as well. This is because a high concentration of bacteria or viruses initially present can result in a reduced or absent signal in the internal control, leading to competition. If both results are negative, it is assumed that the reaction was inhibited or the DNA/RNA extraction was deficient, and the resulting RT-/qPCR data will not be evaluated.

### 4.6. Sequencing

In case of a positive qPCR CPPV-1 result with a Ct-value ≤ 30, the *VP2*-segment was amplified for sequencing according to Pérez et al. [79]. DNA already extracted in a buffer from the intestinal samples served as starting material. The PCR assay was adapted to ReadyMix Taq PCR Reaction Mix (Sigma-Aldrich^®^, P4600, St. Louis, MO, USA) and performed with two amplification sets overlapping the entire VP2-coding region. We used agarose gel electrophoresis to ensure that the amplifications worked. The PCR products with the respective primers were then submitted to Eurofins Genomics^®^ (Ebersberg, Germany) for sequencing. DNASTAR Lasergene^®^ and MEGA7^®^ softwares were used for the assembly, alignment, and analysis of the sequences. Some publications mention amino acid changes to distinguish between CPPV-1 subtypes [17,53]. A ClustalW algorithm was used for alignment, and a phylogenetic tree (based on the nucleotide sequence of *VP2*) was constructed using the maximum likelihood method. The reliability of the obtained phylogenetic tree was assessed using the bootstrap method based on 500 pseudoreplicates.

### 4.7. Statistics and Graphics

We conducted data management and tables with Microsoft^®^ Office Excel 16.63. Maps were designed using QGIS 3.22, with the data on raccoon occurrence in each municipality provided by the Wildlife Research Unit at Agricultural Centre Baden-Wuerttemberg (WFS; LAZBW, Aulendorf, Germany). The statistical software R (version 4.1.2) and RStudio (2022.07.2) were used for the following statistical tests. These were only performed from a sample size ≥10 due to statistical validity. The prevalence, meaning the proportion of raccoons that were PCR-positive for the pathogens examined, and the 95% Clopper–Pearson confidence intervals were calculated for the entire sample and proportionally for the categories age, sex, age-sex cohort, hunting year and area type (rural vs urban). Whether interdependencies exist between the respective categories and a positive result was tested with Fisher’s exact test. In addition, the odds ratios (OR) with 95% confidence intervals were calculated for significant correlations. The Mann–Whitney U (MWU) test was performed to compare whether the body weight and length values differed between negative and positive results. For the statistical tests, a significance level of *p* ≤ 0.05 with a Bonferroni α-adjustment was set, and significant effects are marked with ^§^.

## 5. Conclusions

In summary, the occurrence of pathogens with One Health relevance has been demonstrated in raccoons from BW. The raccoons may participate in the distribution and thus in the environmental pollution of the detected pathogens. They may be a link between wildlife and domestic and zoo animals as well as humans in BW. We conjecture that raccoons indirectly or directly increase the risk of infection of captive, domestic and wild animals with *Ap*, p*L*, CDV, CPPV 1 and humans with the zoonotic pathogens *Ap* and p*L*.

Consequently, it is recommended in areas with high biodiversity, endangered species and humans to keep raccoons away, decimate or eradicate them to minimise their potential negative impact as IAS and as catalysers in part of pathogen pollution [9,15,57,68]. Furthermore, the four cornerstones of prediction, prevention, diagnosis and intervention are necessary in the sense of the One Health approach. Prevention (e.g., hygiene concept, vaccination program, vector control, raccoon management, public education) is essential to reduce the negative effects of the detected pathogens on human and animal health as well as adverse socio-economic effects [66,80,81,82,83].

The presented results underline the importance of wildlife disease surveillance systems which, as part of a One Health approach, take into account the ecology of causative factors such as the impact of biodiversity loss, human impact and the role of invasive species [1,4,5,6]. It is essential to generate knowledge of the role of wildlife in epidemiology, their spreading tendency as well as the negative impact of (non-)zoonotic EIDs and to develop appropriate management measures [1,2,4,5,6,21,28,29]. This exploratory study provides the basis for targeted prevalence studies with larger sample sizes within the area of interest to get a better idea of the infection situation and the infection risk of humans and animals concerning the individual pathogens. Furthermore, similar studies in other German regions and European countries would help to understand the differences in the occurrence of the pathogens [21].

## Figures and Tables

**Figure 1 pathogens-12-00389-f001:**
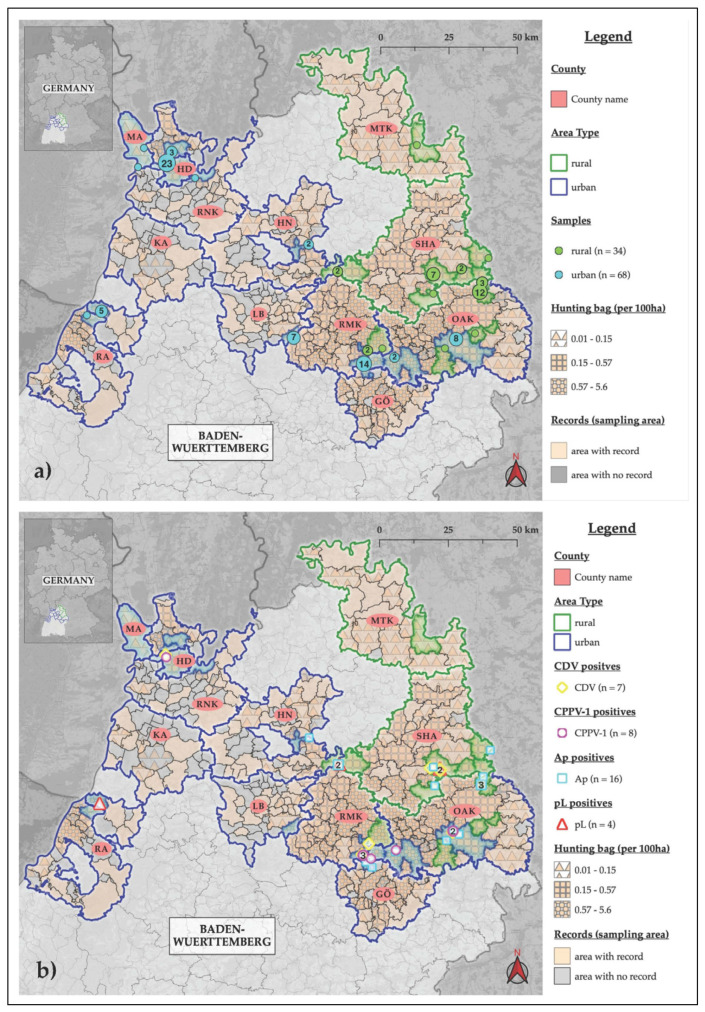
Map of (**a**) raccoon sampling sites and (**b**) pathogen locations. The cumulative illustration of raccoon abundance is based on survey records and hunting bag data from 2019 of the Wildlife Research Unit at Agricultural Centre Baden-Wuerttemberg (WFS; LAZBW, Aulendorf, Germany). The hunting bag describes the number of raccoons hunted per 100 ha. MA (Mannheim), HD (Heidelberg), RNK (Rhein-Neckar-Kreis), RA (Rastatt), KA (Kreis Karlsruhe), HN (Kreis Heilbronn), LB (Kreis Ludwigsburg), RMK (Rems-Murr-Kreis), GÖ (Kreis Göppingen), MTK (Main-Tauber-Kreis), SHA (Kreis Schwäbisch Hall), OAK (Ostalbkreis). (**b**) Canine distemper virus (CDV), carnivore protoparvovirus-1 (CPPV-1), pathogenic *Leptospira* spp. (p*L*), *Anaplasma phagocytophilum* (*Ap*).

**Figure 2 pathogens-12-00389-f002:**
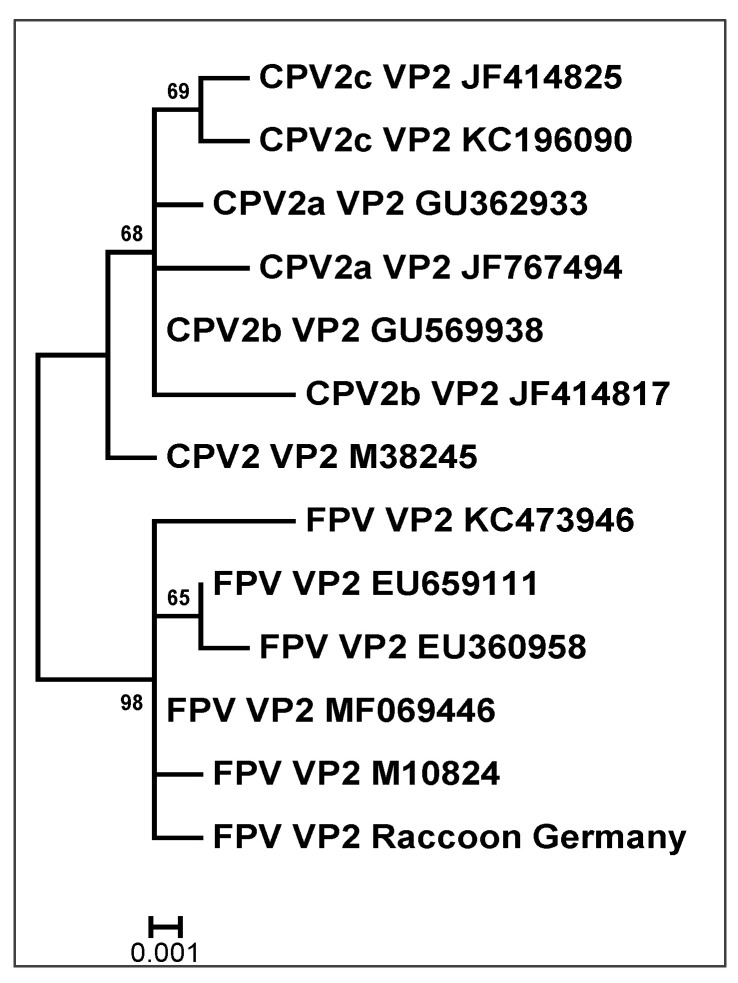
Phylogenetic analysis of FPV_VP2_Raccoon_Germany. ML tree (500 bootstraps) based on the VP2-sequence of the carnivore protoparvovirus-1 (CPPV-1), including reference sequences of the feline panleukopenia virus (FPV) and the canine parvovirus-2 (with subtypes CPV-2a, -2b and -2c). The scale bar indicates nucleotide substitutions per site.

**Table 1 pathogens-12-00389-t001:** Results. The table shows the absolute and relative results with 95% confidence intervals according to the individual categories. The relative frequency describes the proportion of positive results within each category.

			No. Pos (%; CI 95%)			
Parameters	No.	Positives	CPPV-1	CDV	p*L*	*Ap*
**Total**	102	30 (28.3; 20.8–39.3)	8 (7.8; 3.4–14.9)	7 (6.9; 2.8–13.6)	4 (3.9; 1.1–9.7)	16 (15.7; 9.2–24.2)
**Years**						
2019	33 (32.4; 23.4–42.3)	10 (30.3; 15.6–48.7)	1 (3; 1.9–24.3)	3 (9.1; 1.9–24.3)	2 (6.1; 0.7–20.2)	6 (18.2; 7–35.5)
2020	69 (67.6; 57.7–76.6)	20 (29; 18.7–41.2)	7 (10.1; 4.2–19.8)	4 (5.8; 1.6–14.2)	2 (2.9; 0.4–10.1)	10 (14.5; 7.2–25)
**Area**						
rural	34 (33.3; 24.3–43.4)	14 (41.2; 24.6–59.3)	0 (0; 0–10.3)	4 (11.8; 3.3–27.5)	3 (8.8; 1.9–23.7)	10 (29.4; 15.1–47.5) ^§^
urban	68 (66.7; 56.6–75.7)	16 (23.5; 14.1–35.4)	8 (11.8; 5.2–21.9)	3 (4.4; 0.9–12.4)	1 (1.5; 0.03–7.9)	6 (8.8; 3.3–18.2) ^§^
**Sex**						
♀	39 (38.2; 28.8–48.4)	11 (28.2; 15–44.9)	3 (7.7; 1.6–20.9)	4 (10.3; 2.9–24.2)	2 (5.1; 0.6–17.3)	5 (12.8; 4.3–27.4)
♂	63 (61.8; 51.6–71.2)	19 (30.2; 19.2–43)	5 (7.9; 2.6–17.6)	3 (4.8; 1–13.3)	2 (3.2;0.4–11)	11 (17.5; 9.1–29.1)
**Age**						
>1 a	56 (54.9; 44.7–64.8)	23 (41.1; 28.2–55.0)	6 (10.7; 4–21.9)	6 (10.7; 4–21.9)	2 (3.6; 0.4–12.3)	12 (21.4; 11.6–34.4)
<1 a	46 (45.1; 35.2–55.3)	7 (15.2; 6.3–28.9)	2 (4.4; 0.5–14.8)	1 (2.2; 0.1–11.5)	2 (4.4; 0.5–14.8)	4 (8.7; 2.4–20.8)
**Age-Sex Class**						
>1 a, ♀	16 (15.7; 9.2–24.2)	7 (43.8; 19.8–70.1)	2 (12.5; 1.6–38.3)	3 (18.8; 4.1–45.6)	0 (0; 0–20.6)	3 (18.8; 4.1–45.6)
>1 a, ♂	40 (39.2; 29.7–49.4)	16 (40; 24.9–56.7)	4 (10; 2.8–23.7)	3 (7.5; 1.6–20.4)	2 (5; 0.6–16.9)	9 (22.5; 10.8–38.5)
<1 a, ♀	23 (22.5; 14.9–31.9)	4 (17.4; 5–38.8)	1 (4.4; 0.1–21.9)	1 (4.4; 0.1–21.9)	2 (8.7; 1.1–28)	2 (8.7; 1.1–28)
<1 a, ♂	23 (22.5; 14.9–31.9)	3 (13; 2.8–33.6)	1 (4.4; 0.1–21.9)	0 (0; 0–14.8)	0 (0; 0–14.8)	2 (8.7; 1.1–28)

^§^ significant relationships calculated with Fisher’s exact test with a significance level *p* ≤ 0.025. a, year; CPPV-1, carnivore protoparvovirus-1; CDV, canine distemper virus; p*L*, pathogenic *Leptospira* spp.; *Ap*, *Anaplasma phagocytophilum.*

**Table 2 pathogens-12-00389-t002:** Coinfections of positive pathogens.

Raccoon No.	District	Municipality	Area Type	Sex	Age	CDV	*Ap*	CPPV-1	p*L*
WB 18	SHA	Frankenhardt	rural	♂	>1 a	**+**	-	-	**+**
WB 19	SHA	Frankenhardt	rural	♀	<1 a	**+**	-	-	**+**
WB 34	OAK	Aalen	urban	♀	>1 a	**+**	-	**+**	-
WB 73	SHA	Mainhardt	rural	♀	<1 a	-	**+**	-	**+**
WB 91	OAK	Aalen	urban	♂	>1 a	-	**+**	**+**	-

SHA, Schwäbisch Hall; OAK, Ostalbkreis; a, year; CDV, canine distemper virus; *Ap*, *Anaplasma phagocytophilum*; CPPV-1, carnivore protoparvovirus-1; p*L*, pathogenic *Leptospira* spp.; **+**, positive; **-**, negative.

**Table 3 pathogens-12-00389-t003:** Tested sample materials and used qPCR methods.

Pathogen	Sample Material	qPCR
CDV	brain stem, frontal cortex [44,47]	Elia et al. [38]
CPPV-1	small intestine [53]	Decaro et al. [37]
IAV	lung [26,27]	Virotype^®^ Influenza A Kit (Indical)
WNV	brain stem, frontal cortex [77]	Eiden et al. [78]
p*L*	kidney, liver [39]	Adiavet Lepto Real Time Kit^®^ (Bio-X, Adiagene), Ferreira et al. [39]
*Ap*	blood, spleen [40,48]	Courtney et al. [40]

CDV (canine distemper virus), CPPV-1 (carnivore protoparvovirus-1), IAV (influenza A virus), WNV (West Nile virus), p*L* (pathogenic *Leptospira* spp.), Ap (*Anaplasma phagocytophilum*).

## Data Availability

The datasets generated during and/or analysed during the current study are available from the corresponding author on reasonable request.

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
