# Peer review of "Bacterial and Viral Pathogens with One Health Relevance in Invasive Raccoons (Procyon lotor, Linné 1758) in Southwest Germany"

_pathogens, 2023, doi:10.3390/pathogens12030389_

Round 1

Reviewer 1 Report

This manuscript aims to explore the infection status of approximately 100 raccoons examined for 6 different pathogens after opportunistic collection across one state of southern Germany. Pathogen detection was made using qPCR approaches, in some cases followed by Sanger sequencing and phylogenetic analysis.

I found the article well-written but, at the same time, hard to follow. Considering the extensive length of this draft, I was only able to provide with general suggestions:

-       The Introduction is very similar to an extract from a thesis rather than a concise background leading to the research questions as typically done for scientific articles. It could be shortened and made more relevant to the scientific questions this work addresses.

-       The Methods is well-structured. I would specifically reiterate the samples’ origin and their opportunistic collection, in addition to clearly mentioning if ethical approval was needed and, if so, reference the institution and application no. which enabled this work.

-       The Results also read very similarly to a thesis’ extract rather than a scientific article. I would have kept them much more focused on the important findings, which, as this draft stands, become very diluted and difficult to notice. Also be aware of the many tests and P values provided. Have the authors considered the application of a generalised linear model instead a separate different tests to account for the significance of data which may be correlated and not independent?

-       The Discussion is identical to what mentioned above. There are some very interesting results to discuss and I found it generally we-written, but the exciting findings are lost within this extremely long section instead of being properly highlighted.

Reviewer 2 Report

General

This paper describes the screening of raccoons for 6 selected pathogens. Selecting these pathogens as having relevance for One Health is rather easy, as almost all pathogens can be ‘sold’ under this caption. So, the first thing that came to my mind was why were these 6 pathogens selected? When you talk about raccoons and pathogens, often the first one that comes up is Baylisascaris procyonis. Another interesting candidate would be E. multilocularis (fox tapeworm) due to its high prevalence in Baden Württemberg. The raccoon is not a known reservoir of E. multilocularis but little is known and it may actually act as intermediate or dead-end hosts. Hence, a clear rationale why these 6 pathogens were selected is not given. Does the raccoon play a different role for each pathogen selected; reservoir host, propagation host, intermediate host, dead-end host or accidental host? It seems more a random selection. Writing a 4-page introduction section justifying the screening for these 6 pathogens is a little bit too much and the introduction should be written more concisely. The discussion is also very elaborate and without much coherency. The pathogens and the occurrence are discussed separately instead of compared with each other, like linking the comparative occurrence with the ecological behaviour of the raccoon. Now often the same arguments appear in each section (e.g. adaption to urban environment). When it comes to the methodology, it may be helpful to include a table which organs/tissues were investigated for each pathogen as it seems now a more random selection of samples collected and investigated. Also, it is not always clear if the ‘sensitivity’ of the different PCRs were comparable; for example, the fragment size of each PCR used.

The analysis by body weight receives a lot of attention but has only limited value as the authors themselves point out that weight is most likely linked to age. Thus, the positive correlation between weight and infection rate is not very meaningful in this context and could lead to incorrect conclusions.

It seems that a whole dissertation needs to be ‘pressed’ into one publication. The article could be shortened considerably without losing its’ core message. This means major restructuring of the text and omitting redundant and non-essential information/data.

Specific

Line 73: this whole section gives too much detail for the introduction

Line 94: Sentence causes confusion as a result of the word ‘and’. I suggest to replace it with ‘that includes also …’

Line 179-180: This sentence should be restructured as it can cause confusion.

Line 204: Section §2.1.1 – this is just too much data (detail) for a paper on screening raccoons for selected pathogens. It is not really relevant for study purpose. It just makes the paper very long, with the risk that readers will lose interest; one should try to focus on the relevant and such additional data can be moved to a supplementary file.

Line 208-209: Just like the body weight analysis, be careful discussing sex ratio on such a biased sample of predominantly hunted animals.

Line 305: please insert in legend what ‘1’ and ‘0’ means; it is obvious but it should be mentioned

Line 521-522: So, for CPPV-1 lineages were determined but not for CDV. Is there a particular reason?

Line 540: Also, here it is stated that no significant correlation between the detection of CPPV-1 and age, sex, weight, etc. was found, just like with some of the pathogens. The authors clearly state that sample size was too small for the estimation of the prevalence of the pathogens in the raccoon population studied. So, why are you trying to investigate if there is a correlation when you already indicate that sample size is not sufficient for such analysis. You claim it is a screening study and that a valid study objective. Don’t get lost in analysis when the data is not sufficient to do so.

Line 665: it is obvious but a statement like this still needs a reference.

Line 666: have WNV and IA been detected in other wildlife species in BW? Line 673-675, as no IA or WNV in known reservoir species have been detected in the study area/period, it seems strange to look for it in a species like raccoons

Line 704 -705: this sentence could use a reference.

Reviewer 3 Report

Dear Authors, the article titled “What’s in the Box? Exploratory Study of Bacterial and Viral 2 Agents with One Health Relevance in Invasive Raccoons 3 (Procyon Lotor, Linné 1758) in Southwest Germany” is a well written, novel and interesting manuscript.
I recommend publication after minor revisions that I pointed below.

Unfortunately, given the short time granted for the revision process, I was able to complete only a limited part of the revision, apologies about that.

Title can be shortened

Abstract:
line 14: delete “non” and “re” in brackets

Line 19-20: how many positives?

Introduction:

Definitely too long and too detailed.

Line 38-39: delete “non” and “re” in brackets

Personally, I don’t like splitting Introduction section in subchapters, and I ‘m not sure whether editorial policies permits it. Anyway, I recommend the authors in merging the 4 subchapters, maybe shortening some redundant parts.

line 62-72: too detailed description and not relevant for the reader. Can be shortened.

Line 101: change “publications” with “reports” and add reference

Line 134: change “publication” with “report”
Line 135: detected

Line 136-137: not clear the meaning of this sentence. Consider changing “due to negative PCR results” with “due to the low prevalence in this species”.

Discussion:
Are CDV, parvovirus, Leptospira vaccines in dog mandatory in Germany? If so, how this fact could influence the spread of such pathogens in wildlife?
How many stray dogs (that could potentially be the link between domestic animals and raccoons) are there approximately in the studied area?
These points could be added in the paragraph to help the reader understanding the epidemiological situation in wild carnivores.

Reviewer 4 Report

Manuscript ID: pathogens-2213855

Title: What’s in the Box? Exploratory Study of Bacterial and Viral Agents with One Health Relevance in Invasive Raccoons (Procyon Lotor, Linné 1758) in Southwest Germany

Thank you for this study that I find relevant and well writing. There is only one major point, on my mind about the lack of biological hypothesis in the statistical analysis of the positive raccoons.

Main comment:

Lines 281 – 297: Statistical analysis on positive raccoons have no biological significance. Indeed, tested pathogens are very different as their pathogeny. The highlighted significant links can hide different influences depending on pathogens or can be the result of very diverse situations depending on pathogens. For example, adults are more likely positive because of maternal antibodies, seasonality of vector borne disease….

What is the research question lines 289-297? What is the hypothesis? Why in urban area, your positive raccoons for CDV, parvovirus, leptospira and anaplasma should be heavier and longer?  And not in rural area? 

Statistical analyses are tools for exploring hypotheses that are based on knowledge. The relationships sought here do not appear to be based on obvious or unclear assumptions

Minor comments:

Neozoon: is it an English word? 

Title: I found that “What’s in the Box” is not necessary in the title. An animal is not a box and the study explore only a very little part of the “box”.

Line 53 “depending on the definition“: definition of what?

Line 100 “population dynamics of endangered animals“: which ones?

Line 110 “In Europe and Germany, only one publication of CPPV-1 in raccoons is available”: and then? What is the main points highlighted in this publication?

Line 131 “expansion of vector and vertebrate host“: This is not true. If climate crisis and human practices, lead to the expansion of the vectors, it is not the case for vertebrate host. Birds’ population declines and bird’s geographical repartition is not expanding.

Lines 197-198: “highlights the presence of” instead of “documents the occurrence”.

Round 2

Reviewer 2 Report

General comment(s):
The paper has improved considerably; due to its smaller (consise) size it is much more accessible. A compliment to the authors; it is not always easy deleting information and, even more, certain results. I only have a few minor issue (see below) but I also would suggest that a native English speaker will have a look at the manuscript to iron out the few 'rough spots'

Specific comments (based on revised submitted manuscript):

line 38-39: I don't understand 'despite low genetic diversity' in the context of the sentence.

line 117: Figure 1 - In the main text the genotype of the positive sample is referred to as WB91. In the figure, are WB91 and 'FPV-VP2 Raccoon Germany' the same sample? Please use an identical reference in the figure and main text

Line 192: 'Highest density' - May be better to replace with 'a relatively high density', otherwise you have to identify to what you refer (highest density of BW, Germany or Europe?)

Line 210: I would suggest to somehow remove ', those with an urbanized area share of at least 50%," by placing it in parenthesis and re-formulating it a bit

Line 281-282: I guess the authors want to make clear that the CDV infection is likely the cause of the neurological clinical signs. This could be expressed a bit more clearly.

Line 298: 'samples were less sensitive' - I guess the assay could be less sensitive in kidney samples but not the organ itself.

Reviewer 4 Report

Thank you for taking into account my comments

Author Response

Dear Reviewer and Editor,

Thank you for reviewing our manuscript and for your suggestions for changes.